# Impact of Pelvic Fracture Sites on Fibrinogen Depletion in Patients with Blunt Trauma: A Single-Center Cohort Study

**DOI:** 10.3390/jcm11164689

**Published:** 2022-08-11

**Authors:** Mayuko Kunii, Shunichiro Nakao, Yuko Nakagawa, Junya Shimazaki, Hiroshi Ogura

**Affiliations:** Department of Traumatology and Acute Critical Medicine, Osaka University Graduate School of Medicine, Suita 565-0871, Japan

**Keywords:** pelvic fracture, fracture site, blunt injury, coagulopathy, fibrinogen

## Abstract

Background: We aimed to examine the association of pelvic fracture sites with the minimum fibrinogen level within 24 h after hospital arrival. Methods: We conducted a single-center cohort study using health records review. We included patients with pelvic fractures transported by ambulance to a tertiary-care hospital from January 2012 to December 2018 and excluded those transported from other hospitals or aged younger than 16 years. The pelvic fracture was diagnosed and confirmed by trauma surgeons and/or radiologists. We classified the fracture sites of the pelvis as ilium, pubis, ischium, acetabulum, sacrum, sacroiliac joint diastasis, and pubic symphysis diastasis, and each side was counted separately except for pubic symphysis diastasis. We performed linear regression analysis to evaluate the association between pelvic fracture sites and the minimum fibrinogen level within 24 h of arrival. Results: We analyzed 120 pelvic fracture patients. Their mean age was 47.3 years, and 69 (57.5%) patients were men. The median Injury Severity Score was 24, and in-hospital mortality was 10.8%. The mean minimum fibrinogen level within 24 h of arrival was 171.4 mg/dL. Among pelvic fracture sites, only sacrum fracture was statistically significantly associated with the minimum fibrinogen level within 24 h of arrival (estimate, −34.5; 95% CI, −58.6 to −10.4; *p* = 0.005). Conclusions: Fracture of the sacrum in patients with pelvic fracture was associated with lower minimum fibrinogen levels within 24 h of hospital arrival and the requirement of blood transfusion.

## 1. Introduction

Trauma is a leading cause of death globally, accounting for 9% of the world’s deaths [1]. Hemorrhagic shock is the second-leading cause of early death following central nervous system injury [2]. Coagulopathy in the early phase in trauma patients can be caused by hemodilution due to resuscitation, acidosis, hypothermia, shock, and other factors [3]. In patients with severe trauma, the process of coagulopathy begins within a few minutes after injury, and previous studies reported 25–35% of trauma patients had already developed coagulopathy on hospital arrival [4,5,6]. Coagulopathy can result in increased transfusion volume and mortality [7].

Pelvic fracture occurs in about 5–9% of blunt trauma patients [8,9]. The mortality rate varies from 5 to 60% and is especially high in patients with hemodynamic instability [10]. There are various forms of pelvic fracture, and fracture sites may affect the severity of coagulopathy and the amount of blood required for transfusion [11]. Although previous studies suggested that the amount of pelvic hematoma and vascular injuries were more related to the prognosis than to the morphology of pelvic fracture, the amount of hematoma can change as time passes after injury [9,11]. As a previous article mentioned that most pelvic hemorrhage occurs from venous and fracture sites (85%), we hypothesized that the fracture site of pelvis would affect the total amount of bleeding, which may impact on exacerbation of coagulopathy [12]. Pelvic fracture sites can be evaluated by simple imaging tests such as plain computed tomography (CT) and may be related with the amount of hematoma and coagulopathy. Therefore, in this study, we focused on fracture morphology to evaluate its relationship with coagulopathy in blunt trauma patients with pelvic fracture. Fibrinogen is the first coagulation factor to fall below critical levels in trauma patients. Plasma fibrinogen levels decrease earlier and more frequently than other coagulation factors, predicting massive bleeding and death [13]. Previous studies have shown that a low level of fibrinogen is associated with poor outcome, and guidelines for trauma management recommend repeated fibrinogen measurements to monitor coagulation [14,15,16]. The purpose of this study was to examine the association of pelvic fracture sites with the minimum fibrinogen level within 24 h after hospital arrival and the requirement of blood transfusion.

## 2. Materials and Methods

### 2.1. Study Design

We conducted a single-center retrospective cohort study using health records review. This study was approved by the ethics committee of the Osaka University Hospital (approval no.: 20540), which waived the need for informed consent.

### 2.2. Setting

The study was conducted at a tertiary-care hospital in Japan. Our department has 19 intensive care unit beds and treated 285 trauma patients in 2018. Trauma surgeons, neurosurgeons, orthopedic surgeons, and specialists in interventional radiology are available at any time. A dual-source 64-slice CT system (SOMATOM Definition Flash, Siemens Medical Solutions, Forchheim, Germany), a dedicated angiography room, and radiology technicians are also always available in our department.

We administer blood transfusion to patients with signs of hemodynamic instability or excessive bleeding in accordance with the Japanese trauma guideline, which recommends a target hemoglobin level of 7–9 g/dL (70–90 g/L), fibrinogen level of 150–200 mg/dL (1.5–2 g/L), and platelet count of 50,000/µL (50 × 109/L) or 100,000/µL (100 × 109/L) in patients with traumatic brain injury [17].

### 2.3. Participants

We included blunt trauma patients who had pelvic fractures as a discharge diagnosis and were transported by ambulance to the hospital from January 2012 to December 2018. Patients with pelvic fractures were selected by the diagnosis in the patients’ health records. Diagnosis of pelvic fracture was made with CT images confirmed by trauma surgeons and/or radiologists. We excluded patients who were transferred from other hospitals, patients aged 15 years or younger, and patients in whom the fibrinogen level was not measured within 24 h after arrival.

### 2.4. Variables

We extracted the following patient data from the health records: age, sex, mechanism of injury (traffic collisions including car crashes, motorcycle crashes, bicycle crashes, and collisions with pedestrians; fall from height; unknown), Injury Severity Score (ISS), vital signs on hospital arrival, concomitant injury (head, thoracic, abdominal trauma), extravasation on contrast-enhanced CT, angiography, transcatheter arterial embolization (TAE), pelvic gauze packing, length of hospital stay, and in-hospital mortality. Patients with a systolic pressure below 80 mmHg were defined as being in shock.

The pelvic fracture was diagnosed and confirmed by trauma surgeons and/or radiologists. We classified the fracture sites of the pelvis as ilium, pubis, ischium, acetabulum, sacrum, sacroiliac joint diastasis, and pubic symphysis diastasis, and each side was counted separately except for pubic symphysis diastasis. The sacral fractures were also classified according to the Denis classification (Zone I, II, and III) [18]. The fracture site of the pelvis was determined according to the health records reviewed by two study investigators independently, and any discrepancy was resolved by consensus of three investigators. We classified pelvic fractures using Young−Burgees classification (Antero-Posterior Compression APC; Lateral Compression LC; Vertical Shear vs. and Combined Mechanisms CM) and World Society of Emergency Surgery (WSES) classification (grade I to IV) [19].

The primary outcome was minimum fibrinogen level within 24 h of arrival. The secondary outcome was the necessity of blood transfusion within 72 h from arrival. The initial and minimum fibrinogen levels within 24 h of arrival are shown.

### 2.5. Statistical Analysis

Continuous variables are presented as mean with standard error (SD) or median with interquartile range (IQR), and categorical variables are presented as frequencies and percentages, as appropriate. To investigate the association between pelvic fracture sites and coagulopathy, we performed univariable and multivariable linear regression analyses with the dependent variable set as minimum fibrinogen level within 24 h of arrival. We adjusted for the following covariates in the multivariable analysis: concomitant injuries because multiple injuries may relate to the amount of bleeding; age as an important factor of coagulopathy [20]; and shock on arrival, which has a relationship with coagulopathy [21]. We also examined the association between blood transfusion and pelvic fracture sites with univariable and multivariable logistic regression analyses and calculated crude and adjusted odds ratios (ORs) with confidence intervals (CIs). We also performed logistic regression analyses with WSES classification as a covariate instead of shock on arrival. A two-sided significance level of 0.05 was used for statistical inferences. All statistical analyses were performed using R statistical software (version 3.6.2; R Foundation for Statistical Computing, Vienna, Austria).

## 3. Results

We examined the relationship between pelvic fracture sites and coagulopathy. Patient flow in this study is shown in Figure 1. In total, 167 patients with pelvic fracture were brought to our hospital by ambulance from January 2012 to December 2018. We excluded 4 patients aged 15 years or younger, 32 patients who were transferred from other hospitals, and 11 patients with missing data on fibrinogen levels. We thus analyzed 120 patients with pelvic fracture as the study cohort.

Table 1 shows the characteristics of the patients in this study. Their mean age was 47.3 (SD, 20.1) years, and 69 (57.5%) patients were men. The most frequent mechanism of fracture was traffic collision in 67 (55.8%) patients followed by fall from height in 49 (40.8%) patients. The median ISS was 24 (IQR, 13–41), and 24 patients (20.0%) were in shock on arrival. The most common pelvic fracture site was the pubis in 67 (55.8%) patients followed by the ischium in 53 (44.2%) patients and the sacrum in 53 (44.2%) patients. The most frequent type of pelvic fracture was lateral compression in Young−Burgees classification and the grade I in WSES classification. Concomitant injuries included head injury in 47 (39.2%) patients, thoracic injury in 65 (54.2%) patients, and abdominal injury in 30 (25.0%) patients. Thirty-eight (31.7%) patients underwent angiography, and TAE was performed in 34 (28.3%) of them. Pelvic gauze packing was performed in eight (6.7%) patients. The median length of hospital stay was 22 (IQR, 5.75–39) days, and in-hospital mortality was 10.8%.

Table 2 shows the results of fibrinogen level measurements and the requirement of blood transfusion. The mean initial fibrinogen level was 211.7 mg/dL, and the mean minimum fibrinogen level within 24 h of arrival was 171.4 mg/dL. Fifty-nine patients (49.2%) received blood transfusion, among whom 59 (49.2%) received red cell concentrate, 47 (39.2%) received fresh frozen plasma, and 27 (22.5%) received platelet concentrate.

Table 3 shows the results of the univariable and multivariable linear regression analyses assessing the effect of the pelvic fracture site on the minimum fibrinogen level within 24 h of arrival. Only sacrum fracture was statistically significantly associated with the minimum fibrinogen level in the multivariable analysis (estimate, −34.5; 95% CI, −58.6 to −10.4; *p* = 0.005). The results of logistic regression analyses with WSES classification as a covariate instead of shock on arrival were similar as described in the Appendix A. Table 4 shows the results of univariable and multivariable logistic regression analyses assessing the association between pelvic fracture site and the requirement of blood transfusion. Again, only sacrum fracture was associated with an increased risk for the requirement of blood transfusion in the multivariable analysis (adjusted OR, 1.93; 95% CI, 1.08 to 3.59; *p* = 0.031).

## 4. Discussion

We analyzed 120 patients with pelvic fractures and found that fractures of the sacrum were associated both with lower minimum fibrinogen level within 24 h of hospital arrival and with the increased risk for the requirement of blood transfusion.

In this study, the most common mechanism of injury causing pelvic fractures was traffic collision. This result is consistent with a previously reported epidemiology of pelvic fractures [8]. The in-hospital mortality in this study was 10.8%. The mortality from pelvic fractures in previous reports ranges from 5 to 60% depending on the extent of fracture, hemorrhage, and complications, which is consistent with our result [10]. Previous articles reported that 3–20% of pelvic fractures were complicated by arterial injury [22,23,24]. However, 28.3% of the patients with pelvic fracture underwent TAE in this study, which was more frequent than surgical intervention. This may be because we conducted this study in a tertiary-care center with a dedicated angiography room.

There are three main causes of bleeding from pelvic fractures: arterial injury, venous injury, and bleeding from the bone [25]. The most commonly injured arteries in pelvic fracture are the internal iliac artery system, including the superior gluteal artery and external pudendal artery. Bleeding from veins and bone in pelvic fracture can be stopped by the tamponade effect in the retroperitoneal space of the pelvis, but bleeding from arterial injury overcomes the tamponade effect by the surrounding hematoma and more likely causes shock [26]. In this study, the fibrinogen level was significantly decreased in sacrum fracture. This is understandable for the following reasons. Fractures of the sacrum often cause injury of the superior gluteal artery that runs along the anterior surface of the sacrum [27,28]. Injury of the venous plexus on the anterior surface of the sacrum can cause massive bleeding, which is sometimes observed in surgical rectal procedures [29,30]. In addition, injury of the Batson venous plexus, also known as the internal vertebral venous plexus, which runs within the sacral spinal canal and extends from the occipital bone to the coccyx, can also lead to massive bleeding, especially in Denis type III fractures [31]. These anatomical features suggest that sacral fractures can cause concurrent bleeding from both vessels and bone.

In addition, traumatic shock can easily lead to coagulopathy, and the mortality rate in patients with hemodynamically unstable pelvic fractures was previously reported to be as high as 40% [21,32]. There are multiple mechanisms of decrease in plasma fibrinogen levels including coagulation activation-induced consumption, hyper-fibrino(geno)lysis-induced degradation, and dilution by infusion/transfusion as described in a previous article [13]. In this study, fibrinogen levels were significantly decreased in patients with shock on arrival, which is compatible with a traumatic coagulopathy process [4,33].

Unlike thoracic and abdominal trauma, retroperitoneal hemorrhage with pelvic fracture can be difficult to detect with ultrasonography, and it is sometimes difficult to predict the progression of bleeding and coagulopathy in pelvic fracture after performing imaging exams [19]. Our results suggest that when a sacral fracture is found in patients with pelvic fracture, healthcare providers may need to pay more attention to the possible fibrinogen depletion and the requirement of blood transfusion. Our findings might help in predicting the development of coagulopathy from the morphology of the pelvic fracture and the need for early administration of blood transfusions.

Our study has some limitations. First, this was a single-center retrospective study, which raises the issue of selection bias. The results may be different in patients with different severity of trauma or in different medical systems. However, transfusion was provided according to the Japanese guidelines for trauma, which used the same thresholds as in other guidelines [14,17]. Second, not all of the diagnoses of fracture site were made by radiologists. However, the images without a radiology report were diagnosed by multiple trauma surgeons and confirmed by at least two study investigators. Therefore, we believe that any diagnostic errors are limited. Third, we did not include the information on the volume of hematoma or extravasation on CT scan as well as the degree of bone displacement, which may affect the fibrinogen depletion. Instead, we adjusted the results with shock on arrival as well as WSES classification to take the hemodynamic stability into account. Because of the retrospective nature of this study, the minimum fibrinogen levels might have been modified by the administration of fresh frozen plasma, which would cause underestimation of coagulopathy. Furthermore, because repeated measurements of the fibrinogen level were taken at the discretion of treating physicians, the minimum fibrinogen level could be underestimated. However, despite these inevitable underestimations of coagulopathy, our results for the minimum fibrinogen level and the requirement for transfusion were consistent. In addition, treating sacral fractures as one category may be too generalized. Although the Denis classification was used for the sacral fractures, we did not use this classification for the main analysis because the sample size was small, and this classification is often used to identify risks of neurologic injury. Further study is needed to evaluate the association between the fibrinogen depletion and the patterns of sacral fractures.

## 5. Conclusions

Fracture of the sacrum in patients with pelvic fracture was associated with lower minimum fibrinogen levels within 24 h of arrival and the requirement of blood transfusion. Patients with sacral fractures may require extra attention regarding possible fibrinogen depletion.

## Figures and Tables

**Figure 1 jcm-11-04689-f001:**
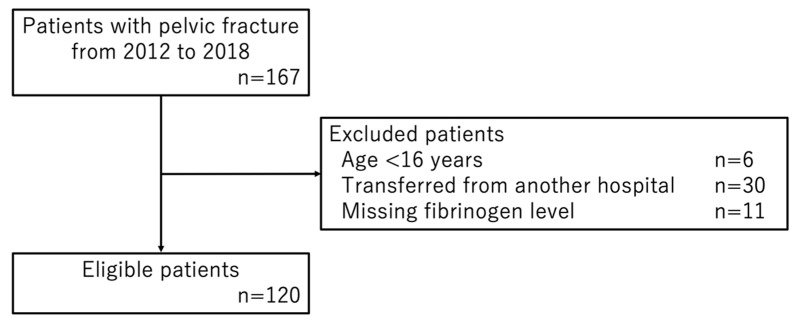
Patient flow.

**Table 1 jcm-11-04689-t001:** Patient characteristics of patients with pelvic fracture.

Characteristics	*n* = 120	
Age, years, mean (SD)	47.3	(20.1)
Male sex, *n* (%)	69	(57.5)
Mechanism, *n* (%)		
Traffic collision	67	(55.8)
Car crash	8	(6.7)
Motorcycle crash	34	(28.3)
Bicycle crash	8	(6.7)
Pedestrian	17	(14.2)
Fall from height	49	(40.8)
Others	4	(3.3)
ISS, median (IQR)	24	(13–41)
Shock on arrival, *n* (%)	24	(20.0)
Pelvic fracture site, *n* (%)		
Ilium	39	(32.5)
Unilateral	37	(30.8)
Bilateral	2	(1.7)
Pubis	67	(55.8)
Unilateral	48	(40.0)
Bilateral	19	(15.8)
Ischium	53	(44.2)
Unilateral	40	(33.3)
Bilateral	13	(10.8)
Acetabulum	49	(40.8)
Unilateral	43	(35.8)
Bilateral	6	(5.0)
Sacrum	53	(44.2)
Unilateral	38	(31.7)
Zone I	11	(9.2)
Zone II	21	(17.5)
Zone III	6	(5.0)
Bilateral	15	(12.5)
Zone I	1	(0.8)
Zone II	4	(3.3)
Zone III	10	(8.3)
Sacroiliac joint disruption, *n* (%)	18	(15.0)
Unilateral	14	(11.7)
Bilateral	4	(3.3)
Pubic symphysis diastasis, *n* (%)	3	(2.5)
Young−Burgees classification, *n* (%)		
APC 3	6	(5.0)
LC 1	51	(42.5)
LC2	41	(34.1)
LC 3	5	(4.1)
vs. and CM	17	(14.2)
WSES classification		
Grade I	46	(38.3)
Grade II	38	(31.7)
Grade III	12	(10.0)
Grade IV	24	(20.0)
Concomitant injury, *n* (%)		
Head	47	(39.2)
Thorax	65	(54.2)
Abdomen	30	(25.0)
Extravasation on contrast-enhanced CT, *n* (%)	34	(28.3)
Angiography, *n* (%)	38	(31.7)
Transcatheter arterial embolization, *n* (%)	34	(28.3)
Pelvic gauze packing, *n* (%)	8	(6.7)
Length of hospital stay, days, median (IQR)	22	(5.75–39)
In-hospital mortality, *n* (%)	13	(10.8)

SD—standard deviation; ISS—injury severity score; APC—antero-posterior compression; LC—lateral compression; vs.—vertical shear; CM—combined mechanisms; WSES—World Society of Emergency Surgery; IQR—interquartile range; CT—computed tomography.

**Table 2 jcm-11-04689-t002:** Results of coagulation parameters and requirement of blood transfusion.

	*n* = 120	
Minimum fibrinogen level within 24 h of arrival (mg/dL), mean (SD)	171.4	(88.9)
Initial fibrinogen level (mg/dL), mean (SD)	211.7	(73.0)
Initial fibrinogen degradation products level (mg/L), mean (SD)	154.2	(180.2)
Initial D-dimer level (μg/mL), mean (SD)	42.8	(48.0)
Initial prothrombin time activity (%), mean (SD)	76.5	(22.4)
Initial activated partial thromboplastin time (s), mean (SD)	35.2	(29.0)
Patients who underwent blood transfusion, *n* (%)	59	(49.2)
Red cell concentrate	59	(49.2)
Fresh frozen plasma	47	(39.2)
Platelet concentrate	27	(22.5)

SD—standard deviation.

**Table 3 jcm-11-04689-t003:** Results of univariable and multivariable linear regression analyses assessing pelvic fracture site on minimum fibrinogen level within 24 h of arrival.

	Univariable Analysis			Multivariable Analysis		
	Estimate	SE	*p* Value	Estimate	SE	*p* Value
Pelvic fracture site						
Ilium	9.3	16.0	0.563	7.6	16.1	0.638
Pubis	−23.7	11.1	0.035	0.0	13.6	0.999
Ischium	−20.3	11.8	0.088	−13.9	13.3	0.299
Acetabulum	−7.7	13.8	0.577	−1.4	13.0	0.913
Sacrum	−37.1	11.1	0.001	−34.5	12.2	0.005
Sacroiliac joint diastasis	−34.0	17.2	0.051	−2.7	21.8	0.902
Pubic symphysis diastasis	−37.7	52.1	0.470	6.3	58.7	0.915
Concomitant injury						
Head	−27.8	16.5	0.095	−34.7	18.9	0.069
Thorax	−45.6	15.8	0.005	−20.1	16.8	0.234
Abdomen	−27.9	18.6	0.137	−19.0	18.9	0.316
Age	0.0	0.4	0.997	0.0	0.4	0.977
Shock on arrival	−70.2	19.3	<0.001	−46.1	20.9	0.030

SE—standard error.

**Table 4 jcm-11-04689-t004:** Results of univariable and multivariable logistic regression analyses assessing the association between pelvic fracture site and requirement of blood transfusion.

	Crude OR	95% CI	*p* Value	Adjusted OR	95% CI	*p* Value
Ilium	1.7	0.8–3.5	0.171	2.1	0.9–5.1	0.101
Pubis	2.1	1.2–3.5	0.008	1.4	0.7–2.8	0.317
Ischium	1.9	1.1–3.4	0.025	1.5	0.8–3.0	0.236
Acetabulum	1.0	0.5–1.8	0.990	1.0	0.5–2.0	0.987
Sacrum	2.0	1.2–3.4	0.015	1.9	1.1–3.6	0.031
Sacroiliac joint dissection	3.0	1.3–9.2	0.025	2.1	0.6–8.4	0.262
Pubic symphysis dissection	2.1	0.2–46.0	0.548	0.4	0.0–12.1	0.532

OR—odds ratio; CI—confidence interval.

## Data Availability

Data can be available upon request. Please contact the corresponding author to receive information on the dataset of this study.

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
