# Peer review of "Impact of Pelvic Fracture Sites on Fibrinogen Depletion in Patients with Blunt Trauma: A Single-Center Cohort Study"

_jcm, 2022, doi:10.3390/jcm11164689_

Round 1
Reviewer 1 Report
I am grateful to the editors of the JCM journal for the opportunity to review the results of the study of Mayuko Kunii et al. "Impact of pelvic sites fracture on fibrinogen depletion in patients with blunt trauma: a single-center cohort study".
The authors were able to demonstrate that sacral fractures are associated with minimum fibrinogen levels. At the same time, the methods used to assess coagulopathy, in terms of fibrinogen consumption or depletion in pelvic fractures, and the impact of this condition on the treatment strategy and methods, as well as on disease outcomes require substantial revision.
The authors should indicate the exact inclusion and exclusion criteria and use the international classification of pelvic injuries, from APC or even better WSES, which is based on hemodynamic and mechanical stability. This would make it possible to more clearly form groups, to understand which patients have indication for immediate blood or plasma transfusion, embolization or tamponade, and which are amenable for additional examination (i.e. CT). The authors focused only on fibrinogen values, not taking into account other parameters of coagulation panel. I could continue this list of shortcomings, but this is not the main thing.
The main question is how the study results will impact on the strategy and methods for treating pelvic fractures. If a patient with the APC type II or III fracture and WSES grade IV has normal fibrinogen levels, does he/she not require a transfusion of packed RBCs and fresh frozen plasma? Or vice versa, will a patient with APC type I and WSES grade I with low fibrinogen level need blood and plasma transfusion? Of course not, in the settings of a pelvic fracture, which is most often combined with injuries of other abdominal and thoracic organs, head and urogenital tract, the fibrinogen levels are not significant, in my opinion. Pulse, pressure, red blood cell count, hematocrit, signs of hemoperioneum, trauma to the liver, spleen or kidney, fracture instability, etc. – these are determinants influencing decision-making algorithm and guiding the surgeon’s behavior. The authors indicate that embolization and tamponade were used to treat patients. What were the associations between the use of arterial embolization or extraperitoneal pelvic packing and fibrinogen levels? Did the fibrinogen level influence the performance of these interventions? Probably not. Well, the fact is that with massive bleeding from the sacral venous plexus, fibrinogen level will decrease. Yes, this is natural, as fibrinogen is actively used in such cases and its “reserves” are depleted. Moreover, in moderate to severe pelvic injuries, plasma transfusion is one of the most important components of treatment, and hemostasis of utmost importance for successful treatment.
Was there any association between fibrinogen levels and 24-hour mortality? And can this parameter be considered as a factor of an unfavorable outcome?
In general, it seems that authors presented only a part of their great work. Much remains outside the scope of this manuscript, and the authors should continue their research, supplementing the work with additional information.
Author Response
Reviewer 1
I am grateful to the editors of the JCM journal for the opportunity to review the results of the study of Mayuko Kunii et al. "Impact of pelvic sites fracture on fibrinogen depletion in patients with blunt trauma: a single-center cohort study".
The authors were able to demonstrate that sacral fractures are associated with minimum fibrinogen levels. At the same time, the methods used to assess coagulopathy, in terms of fibrinogen consumption or depletion in pelvic fractures, and the impact of this condition on the treatment strategy and methods, as well as on disease outcomes require substantial revision.
The authors should indicate the exact inclusion and exclusion criteria and use the international classification of pelvic injuries, from APC or even better WSES, which is based on hemodynamic and mechanical stability. This would make it possible to more clearly form groups, to understand which patients have indication for immediate blood or plasma transfusion, embolization or tamponade, and which are amenable for additional examination (i.e. CT).
Thank you very much for your suggestion. Inclusion and exclusion criteria were as described in the methods section, regardless of the type of pelvic fracture. We added the internationally recognized Young-Burgees classification and World Society of Emergency Surgery classification to understand the patient characteristics. We added the following sentences in the methods and results sections, and revised Table 1.
“We classified pelvic fractures using Young-Burgees classification (Antero-Posterior Compression APC; Lateral Compression LC; Vertical Shear VS and Combined Mechanisms CM) and World Society of Emergency Surgery (WSES) classification (grade I to IV) [18].” (page 3, para 1)
“The most frequent type of pelvic fracture was lateral compression in Young-Burgees classification and the grade I in WSES classification.” (page 3, para 5)
We also performed multivariable logistic regression analysis with WSES classification instead of shock on hospital arrival as WSES classification can express the severity of pelvic fracture from mild to severe (grade I to IV), and the results were the similar to the initial analysis. We added the following sentences in the methods and results sections, and described the results in a supplementary table.
“We also performed logistic regression analyses with WSES classification as a covariate instead of shock on arrival.” (page 3, para 3)
“The results of logistic regression analyses with WSES classification as a covariate instead of shock on arrival were similar as described in Supplementary Table.” (page 4, para 1)
The authors focused only on fibrinogen values, not taking into account other parameters of coagulation panel. I could continue this list of shortcomings, but this is not the main thing.
We chose the minimum fibrinogen levels within 24 hours from arrival as the primary outcome because fibrinogen level in trauma patients were known as an important predicting factor for mortality. To clarify this, we added the following sentence in the introduction section.
“Plasma fibrinogen levels decrease earlier and more frequently than other coagulation factors, predicting massive bleeding and death [13].” (page 2, para 1)
The main question is how the study results will impact on the strategy and methods for treating pelvic fractures. If a patient with the APC type II or III fracture and WSES grade IV has normal fibrinogen levels, does he/she not require a transfusion of packed RBCs and fresh frozen plasma? Or vice versa, will a patient with APC type I and WSES grade I with low fibrinogen level need blood and plasma transfusion? Of course not, in the settings of a pelvic fracture, which is most often combined with injuries of other abdominal and thoracic organs, head and urogenital tract, the fibrinogen levels are not significant, in my opinion.
We appreciate the opportunity to clarify this issue. If a patient with WSES grade IV has normal fibrinogen levels, hemostasis and transfusion would be required regardless of the fibrinogen level. Patients with APC type I and WSES grade I with low fibrinogen level might need plasma transfusion in case with the delayed bleeding from a minor fracture site. A normal fibrinogen level in those with pelvic fracture could decrease below critical level depending on the time from the injury and the volume of bleeding, resulting in requiring transfusion of fresh frozen plasma. We conducted this research as we hypothesized that the fracture site might affect the fibrinogen depletion in pelvic fracture. To address this, we included with WSES classification as an independent variable instead of shock on arrival in the multivariable logistic regression analysis. Although our findings would not impact on the strategy and methods for treating pelvic fractures, we believe that our findings may help categorize the risk of coagulopathy. As the results were similar to the initial analysis, we added the findings in a supplementary table and added the sentences in the methods and results sections as follows.
“We also performed logistic regression analyses with WSES classification as a covariate instead of shock on arrival.” (page 3, para 3)
“The results of logistic regression analyses with WSES classification as a covariate instead of shock on arrival were similar as described in Supplementary Table.” (page 4, para 1)
Pulse, pressure, red blood cell count, hematocrit, signs of hemoperioneum, trauma to the liver, spleen or kidney, fracture instability, etc. – these are determinants influencing decision-making algorithm and guiding the surgeon’s behavior. The authors indicate that embolization and tamponade were used to treat patients. What were the associations between the use of arterial embolization or extraperitoneal pelvic packing and fibrinogen levels? Did the fibrinogen level influence the performance of these interventions? Probably not. Well, the fact is that with massive bleeding from the sacral venous plexus, fibrinogen level will decrease. Yes, this is natural, as fibrinogen is actively used in such cases and its “reserves” are depleted. Moreover, in moderate to severe pelvic injuries, plasma transfusion is one of the most important components of treatment, and hemostasis of utmost importance for successful treatment.
We investigated minimum fibrinogen levels within 24 hours from arrival rather than initial fibrinogen levels on arrival to take into account the possible exacerbation of coagulopathy after arrival. We agree that surgeon would make decision according to multiple factors the reviewer listed. We understand that our results are not enough to influence decision-making algorithm and guide the surgeon’s behavior. Although we may just find natural phenomenon, we believe that our findings may help categorize the risk of coagulopathy.
Was there any association between fibrinogen levels and 24-hour mortality? And can this parameter be considered as a factor of an unfavorable outcome?
We cannot evaluate the association between fibrinogen levels and 24-hour mortality because no patient died within 24 hours in our cohort. We need further study to address this question.
In general, it seems that authors presented only a part of their great work. Much remains outside the scope of this manuscript, and the authors should continue their research, supplementing the work with additional information.
Thank you very much for this encouraging comments. We continue our research to address knowledge gap in pelvic fracture and coagulopathy.
Reviewer 2 Report
Good paper. One suggestion only.
The word "dissection" is not good for describing the traumatic anatomy of the pubis or the sacrum. The proper term is "diastasis" of the pubic symphysis or the sacroiliac joint. Please revise. this occurs in the Abstract, Intro, Methods and Table 1,3.
Author Response
Reviewer 2
Good paper. One suggestion only.
The word "dissection" is not good for describing the traumatic anatomy of the pubis or the sacrum. The proper term is "diastasis" of the pubic symphysis or the sacroiliac joint. Please revise. this occurs in the Abstract, Intro, Methods and Table 1,3.
Thank you very much for your suggestion. We fixed the terminology in the Abstract, Intro, Methods and Table 1,3.
Reviewer 3 Report
Thank you for permitting me to review this manuscript
In this manuscript the authors intended to eamine the association of pelvic fracture sites with the minimum fibrinogen level within 24 52 hours after hospital arrival and the requirement of blood transfusion.Results of univariable and multivariable linear regression analyses assessing pelvic fracture on minimum fibrinogen level is not conclusive and there is no specificity in the fracture site except for sacrum sites fracture.
here are my concerns
The authors should explain why different sites fracture would have different set of coagulopathy justifying the research
in addition the decrease of fibrinogen after trauma is not a new information here is one example of such research and it could be inserted in the references list and discussed
Hayakawa Journal of Intensive Care (2017) 5:3 DOI 10.1186/s40560-016-0199-3 Dynamics of fibrinogen in acute phases of trauma Mineji Hayakawa
Methods ; this is a retrospective study , it should be clearly stated in the method section , not only in the discussion section
the fibrinogen depletion is mostly related to blood loss
was there any method to assess blood loss for ex via the Ct scan ? may be the sacrum fractures bleed more may be not , but common sense may indicate the amount for bone displacement is proportionnal to final bleeding and consequently to the decrease in fibrinogen level
Author Response
Reviewer 3
Thank you for permitting me to review this manuscript
In this manuscript the authors intended to eamine the association of pelvic fracture sites with the minimum fibrinogen level within 24 52 hours after hospital arrival and the requirement of blood transfusion.Results of univariable and multivariable linear regression analyses assessing pelvic fracture on minimum fibrinogen level is not conclusive and there is no specificity in the fracture site except for sacrum sites fracture.
here are my concerns
The authors should explain why different sites fracture would have different set of coagulopathy justifying the research
We thank the opportunity to clarify this point. As a previous article reported by Kim, et al. mentioned that most pelvic hemorrhage occurs from venous and fracture sites (85%), we hypothesized that the fracture site of pelvis would affect the total amount of bleeding, which may impact on exacerbation of coagulopathy. To clarify this, we added the following statement in the introduction section:
“As a previous article mentioned that most pelvic hemorrhage occurs from venous and fracture sites (85%), we hypothesized that the fracture site of pelvis would affect the total amount of bleeding, which may impact on exacerbation of coagulopathy [12].” (page 1, para 2)
in addition the decrease of fibrinogen after trauma is not a new information here is one example of such research and it could be inserted in the references list and discussed
Hayakawa Journal of Intensive Care (2017) 5:3 DOI 10.1186/s40560-016-0199-3 Dynamics of fibrinogen in acute phases of trauma Mineji Hayakawa
Thank you for your suggestion. We added the following sentence in the discussion section.
“There are multiple mechanisms of decrease of plasma fibrinogen levels including coagulation activation-induced consumption, hyper-fibrino(geno)lysis-induced degradation, and dilution by infusion/transfusion as described in a previous article [13].” (page 6, para 4)
Methods ; this is a retrospective study , it should be clearly stated in the method section , not only in the discussion section
We added “retrospective” in the Study design of the methods section as suggested. (page 2, para 2)
the fibrinogen depletion is mostly related to blood loss
was there any method to assess blood loss for ex via the Ct scan ? may be the sacrum fractures bleed more may be not , but common sense may indicate the amount for bone displacement is proportionnal to final bleeding and consequently to the decrease in fibrinogen level
Unfortunately, we did not include the information on the volume of hematoma or extravasation on CT scan as well as the degree of bone displacement. However, we additionally assessed the multivariable analysis with WSES classification, which indicate the severity of pelvic fracture, and the results were similar to the initial analysis. We added this issue in the limitation section as follows.
“Third, we did not include the information on the volume of hematoma or extravasation on CT scan as well as the degree of bone displacement, which may affect the fibrinogen depletion. Instead, we adjusted the results with shock on arrival and WSES classification to take the hemodynamic stability into account.” (page 7, para 2)
We also added the following sentences in the methods and results sections.
“We also performed logistic regression analyses with WSES classification as a covariate instead of shock on arrival.” (page 3, para 3)
“The results of logistic regression analyses with WSES classification as a covariate instead of shock on arrival were similar as described in Supplementary Table.” (page 4, para 1)
Round 2
Reviewer 1 Report
Unfortunately, my opinion has not changed. The presented work has low scientific and practical significance.
Author Response
Although the reviewer mentioned that we presented only a part of our work, all the relevant data in our study were already presented. As we collected other coagulation parameters that we did not use in the main analysis, we added results of initial FDP, D-dimer, PT activity and aPTT levels in the Table 2. We hope the reviewer understands the limitation of our study and contribution to trauma research.
Reviewer 3 Report
The authors have responded to all my queries
Author Response
We appreciate the reviewer’s work and comments to improve our manuscript.